# Highly Sensitive and Selective Detection of Hydrogen Using Pd-Coated SnO_2_ Nanorod Arrays for Breath-Analyzer Applications

**DOI:** 10.3390/s22052056

**Published:** 2022-03-07

**Authors:** Hwaebong Jung, Junho Hwang, Yong-Sahm Choe, Hyun-Sook Lee, Wooyoung Lee

**Affiliations:** 1Department of Materials Science and Engineering, Yonsei University, 50 Yonseiro, Seodaemun-gu, Seoul 03722, Korea; interfd@yonsei.ac.kr (H.J.); turtlesoup@yonsei.ac.kr (J.H.); 2Isenlab Inc., Halla Sigma Valley, Dunchon-daero 545, Jungwon-gu, Seongnam-si 13215, Korea; cys@isenlab.com

**Keywords:** breath hydrogen analyzer, gas sensor, Pd-coated SnO_2_ nanorods, gas chromatography, selectivity

## Abstract

We report a breath hydrogen analyzer based on Pd-coated SnO_2_ nanorods (Pd-SnO_2_ NRs) sensor integrated into a miniaturized gas chromatography (GC) column. The device can measure a wide range of hydrogen (1–100 ppm), within 100 s, using a small volume of human breath (1 mL) without pre-concentration. Especially, the mini-GC integrated with Pd-SnO_2_ NRs can detect 1 ppm of H_2_, as a lower detection limit, at a low operating temperature of 152 °C. Furthermore, when the breath hydrogen analyzer was exposed to a mixture of interfering gases, such as carbon dioxide, nitrogen, methane, and acetone, it was found to be capable of selectively detecting only H_2_. We found that the Pd-SnO_2_ NRs were superior to other semiconducting metal oxides that lack selectivity in H_2_ detection. Our study reveals that the Pd-SnO_2_ NRs integrated into the mini-GC device can be utilized in breath hydrogen analyzers to rapidly and accurately detect hydrogen due to its high selectivity and sensitivity.

## 1. Introduction

Humans are hosts to over 1000 species of microbes [1] that are responsible for human health and disease [2]. Important and promising studies have identified the effects of the human microbiome on disease conditions, such as obesity [3,4], diabetes [5,6], irritable bowel syndrome [7,8], fatty liver [9], and many others [10,11]. The majority of the microbes in the body inhabit the gastrointestinal tract, giving a chance for these organisms to influence metabolism, immunity, and digestion [12]. It is known that the gastrointestinal tract produces various gases, such as hydrogen, carbon dioxide, methane, and hydrogen sulfide, as the products of intestinal microbial metabolism, colonization, and subsequent fermentation [12]. Most of the gases are absorbed through the lining of the large intestine into the bloodstream. Then, the gases are transported to the lungs via the bloodstream where they are exchanged into the airways of the lungs and exhaled [13].

The source of H_2_ in the breath can be from the bacterial fermentation of carbohydrates in the bowel caused by gastrointestinal diseases. Patients with gastrointestinal disorders have more aggressive fermentation, which results in premature and excessive amounts of H_2_ due to inordinate microbial colonization [14]. The phenomenon is known as small intestinal bacterial overgrowth (SIBO). Therefore, some symptoms of irritable bowel syndrome (IBS) can be described by SIBO, and it can be diagnosed by the hydrogen breath test. However, the hydrogen breath test with lactulose can only diagnose one-third of the patients with SIBO, and the glucose hydrogen breath test is well-known to be highly specific for SIBO diagnoses [14]. On the other hand, some patients with IBS produce gases, such as methane and hydrogen sulfide with or without hydrogen. In particular, patients with constipation may have higher methane contents, whereas patients with diarrhea, such as those with ulcerative colitis and Crohn’s disease, may have higher hydrogen content [14,15]. Thus, hydrogen and methane have the potential for use as biomarkers for diarrhea and constipation, respectively. Accordingly, hydrogen breath tests can be useful for diagnosing many gastroenterological disorders, such as lactose and fructose malabsorption [16], celiac disease [10], Crohn’s disease, and ulcerative colitis [15].

Relatively accurate hydrogen concentration measurements in the breath can be realized using various instrumentation. Nielsen et al. first introduced chromatographic techniques for quantitatively measuring small amounts of H_2_ and CH_4_ in breath samples and applied this method to humans [17,18]. Based on Nielsen’s approach, various detectors, such as gas chromatography (GC) integrated with solid-state sensors [15,19,20], an infrared sensor combined with an electrochemical sensor [21], and a pulsed-discharge helium ionization detector (PHID) coupled with GC [22,23], have been developed to measure H_2_ and CH_4_ in breath samples. Among them, Raman gas spectroscopy exhibits excellent chemical selectivity for various gases in complex mixtures [24,25]. However, most instruments for measuring breath hydrogen and methane are still expensive, not portable, and complex. It is, therefore, necessary to develop such instruments with inexpensive and miniaturized components and simplified detection procedures. 

To develop a compact analyzer, it is critical to find high-performance sensing materials. Metal-oxide-semiconductor (MOS) based sensors can be used in a portable gas sensing system owing to several of their advantages, including high-sensitivity, fast response-time, compact size, and low cost. Specifically, it is easy to fabricate nanostructures to achieve high-sensitivity [26,27,28,29,30,31]. Many metal oxides, including SnO_2_, ZnO, WO_3_, TiO_2_, In_2_O_3_, Nb_2_O_5_, FeO, NiO, Fe_2_O_3_, Ga_2_O_3_, MoO_3_, Sb_2_O_5_, and V_2_O_5_, which exhibit a large variation in electrical resistance after exposure to hydrogen gas, have been investigated for use as hydrogen sensing materials in MOS [27,28]. Among the MOS sensing materials, SnO_2_ has been widely used to detect H_2_ because of its high-sensitivity [29]. Additionally, the hydrogen sensing performance of SnO_2_ has been further enhanced by utilizing noble metals [30]. Notably, Pd is the most widely used metal for enhancing the sensing response of H_2_ and decreasing the optimal working temperature [29,31]. However, most studies conducted on SnO_2_-based hydrogen sensors have focused on detecting H_2_ leakage during storage/transport/use and detecting H_2_ dissolved in transformer oil; it has been found that the lowest detection limit is 4% for the former [32] and 100–500 ppm for the latter [33]. Both results are much higher than that of the concentration of breath hydrogen. The concentration of breath hydrogen is approximately 2 ppm [34] in healthy people and greater than 10 ppm [35] in patients with carbohydrate malabsorption. Therefore, research concerning SnO_2_-based hydrogen sensors for the application of the exhaled breath analyzer is valuable. Furthermore, selective detection of target gas in a mixed gas is a significant obstacle in MOS sensors such as those based on SnO_2_. Consequently, it is necessary to adopt GC to resolve the selectivity issue of MOS sensors.

In this work, we investigated the sensing performance of Pd-coated SnO_2_ nanorod arrays (NRs) integrated with a miniaturized GC column to detect H_2_ selectively for an application of a breath hydrogen analyzer. The H_2_ sensing properties of the breath analyzer were investigated based on the sensing responses at various concentrations of H_2_ (1–100 ppm). The selectivity of the breath analyzer for detecting H_2_ was studied by exposing it to mixtures of gases, such as H_2_, CO_2_, N_2_, CH_4_, and CH_3_COCH_3_. We compared the Pd-SnO_2_ NRs with ZnO nanoparticles, which lack selectivity in detecting H_2_. We also discuss the humidity effect on the sensing performance of the breath hydrogen analyzer.

## 2. Materials and Methods

### 2.1. Fabrication and Characterization

The vertically oriented SnO_2_ nanorods (NRs) were synthesized on an Al_2_O_3_ substrate (5 × 2.5 mm^2^) by the glancing angle deposition method (GLAD) using an electron-beam evaporator. In the GLAD method, by adjusting the incident angle of the vapor flux and the substrate rotation speed, vertically ordered nanorod arrays with high-density, as well as nanorods of uniform diameter and length can be fabricated [31]. Before the synthesis of the NRs, a DC sputtering system was used to fabricate interdigitated electrodes (Pt, 100 nm in thickness) and a heater (Pt, 600 nm in thickness) on the top and bottom of the alumina substrate, respectively. The patterned alumina substrate was loaded onto the sample holder in the sputtering system. SnO_2_ granules (99.99% Kojundo Chemical Laboratory Co., Ltd., Saitama, Japan) that had an average size of 3 mm were placed in a crucible that was located 50 cm below the sample holder. The sample holder was tilted at 80° and rotated at a speed of 15 rpm during the growth of the SnO_2_ NRs. Electron-beam evaporation was performed with a 1 Å/s deposition rate at a base pressure of 5.0 × 10^−6^ Torr. The as-deposited sample was annealed at 550 °C for 2 h. After the synthesis of the SnO_2_ NRs, a 5 nm Pd layer was deposited on the annealed SnO_2_ NRs with a power of 20 W in an Ar atmosphere using an ultra-high vacuum DC magnetron sputtering system. The base pressure was 4.1 × 10^−8^ Torr, and the deposition process was conducted at a pressure of less than 2.3 × 10^−3^ Torr with an Ar flow rate of 34 SCCM. A high-purity Pd target (4N) was used, and the Pd deposition rate was ~4.7 Å/s at room-temperature. The morphology and the compositional analysis of the Pd-SnO_2_ NRs were examined by a field emission scanning electron microscope (FE-SEM, JSM-7001F, JEOL Ltd., Tokyo, Japan) equipped with an energy dispersive X-ray spectrometer (FE-SEM-EDX, JSM-7001F, JEOL Ltd., Tokyo, Japan).

ZnO NPs also were prepared by using a wet chemical method in order to compare the sensing performance of the Pd-SnO_2_ NRs to the ZnO nanoparticles (NPs) recently developed by our group [36]. The synthesized ZnO NPs were dispersed in methanol. More details of the synthesis method for ZnO NPs are available in our previous work [36]. The sensor was fabricated by dropping the ZnO NPs solution onto the patterned Al_2_O_3_ substrate and heat-treated at 600 °C for 30 min.

### 2.2. Gas Sensing Test 

We conducted an H_2_ sensing test using miniaturized gas chromatography (mini-GC). Figure 1 shows the mini-GC that had the dimensions of 8 cm × 13 cm × 16 cm. The mini-GC gas analyzer consisted of a sensor based on Pd-SnO_2_ NRs, a packed column, a sampling loop (1 mL), a mini-sized pump, and three solenoid valves. The sampling volumes of the gases were limited to 1 mL. We use dry air as a carrier gas to continuously supply oxygen ions to the surface of the Pd-SnO_2_ NRs because the gas sensing mechanism of metal-oxide semiconducting materials, such as SnO_2_, is based on changes in the thickness of the depletion layer by oxygen ion adsorption onto the surface. The flow rate of the carrier gas affects the sensing response because the sensing material requires a certain amount of time to react with the gas. Herein, the flow rate of the carrier gas was adjusted using a flow control valve and maintained 30 SCCM for analysis. Prior to injection into the GC system, the ambient air was passed through a filter filled with silica gel to remove any moisture.

We chose a packed GC column because of its huge surface area, low cost, and good operability at room-temperature [37]. The packed column was filled with a packing material coated with the stationary phase (Isenlab Inc., Seongnam-si, South Korea) and the column’s inner diameter was 0.15 cm. The length of the column was adjusted to 30 cm to achieve a short retention time of the target gas. Gas separation occurs during the mixture of gas passing through the column, due to the differing interaction strengths between the various gas molecules and the stationary phase. The interaction strength is related to the polarity and size of the gas molecules and the stationary phase of the packed column was weakly polar. Thus, small non-polar gases, such as H_2_, N_2_, and CO_2_, were quickly released from the column owing to their weak interactions while larger polar gases, such as acetone, were slowly released owing to their strong interactions with the stationary phase.

The sensing process of the breath hydrogen analyzer based on Pd-SnO_2_ NRs is described as follows. First, a 1 mL mixture of the target gas and ambient air was used to fill the sampling loop for 100 s without pre-concentration. Subsequently, the sampled gas was injected into the packed column and the components of the gas were separated by the interaction between the stationary phase and gas molecules as they passed through in the packed column. The temperature of the column was maintained at room-temperature (30 °C) by controlling the thermostat. The target gas that passed through the column was detected by the Pd-SnO_2_ NR or ZnO NPs. To optimize the working temperature of the sensor, the sensor response was tested at various temperatures, i.e., 25, 80, 115, 139, 152, 165, 178, and 191 °C. The ability of the analyzer to sense the presence of H_2_ was checked at various concentrations of H_2_ in the range of 1–100 ppm at the optimal working temperature. The hydrogen concentration was controlled by changing the mixing ratio between highly concentrated hydrogen in an airbase and synthetic air using a mass flow controller. The selectivity of the mini-GC system for H_2_ was investigated by measuring its responses when exposed to various gases, including H_2_, CO_2_, N_2_, CH_4_, and CH_3_COCH_3_ and comparing its responses to those of ZnO nanoparticles.

The sensing performance was evaluated based on the resistance of the sensor, and the resistance obtained in the gas analyzer was converted to a sensor signal (log(*R*)). The change in the sensing signal of the samples was defined as Δ log(*R*) [38].
Δ log(*R*) = log(*R*)_max_ − log(*R*)_min_,(1)
where log(*R*)_max_ and log(*R*)_min_ are the maximum and minimum values before and after exposure to H_2_, respectively. 

### 2.3. Gas Sensing Mechanism 

Figure 2 shows the underlying sensing mechanism of the Pd-SnO_2_ NRs for the detection of H_2_. When the Pd-SnO_2_ NRs were exposed to air, oxygen molecules in the air were adsorbed on the surfaces of the Pd nanoparticles (NPs) and the SnO_2_ NR. The adsorbed oxygen molecules took the electrons from the conduction band of the SnO_2_ NR and formed anionic oxygen species (O_2_^−^, O^−^, and O^2−^) depending on the working temperature. At temperatures greater than 150 °C, O^−^ and O^2−^ were chemisorbed [39,40]. O_2_^−^ commonly is chemisorbed at low temperatures (T < 150 °C) [41]. Since the working temperature of SnO_2_ NR was in the range of >150 °C, the O^−^, and O^2−^ ions were adsorbed on the surface of the SnO_2_ NRs. In addition, oxygen molecules can be adsorbed on the surface of the Pd NPs and be easily dissociated into O^−^ ions due to the spillover effect of the Pd metal [42]. Due to the higher work function of Pd metal (Φ_Pd_ = 5.12 eV) compared to that of SnO_2_ (Φ_SnO_2__ = 4.9 eV), a Schottky barrier was formed at the interface between Pd and SnO_2_ [43], and the electrons that were associated with SnO_2_ were transferred to the Pd metal. This resulted in the enhanced dissociation of oxygen molecules into O^−^ ions on the surface of the Pd NPs, and these ions easily diffused into the surface vacancies of the SnO_2_ NR. Consequently, thicker electron depletion layers were formed at the interface between the Pd NPs and the SnO_2_ NR. This infers that a greater number of reactive sites for H_2_ was produced on the surface of the Pd-SnO_2_ NR.
(2)$$H_{2}\left(\mathrm{gas}\right)\;\to \;2H\left(\mathrm{adsorb}\right)$$
(3)$$2H\left(\mathrm{adsorb}\right)+O^{-}\left(\mathrm{adsorb}\right)\;\to H_{2}O+e^{-}$$

When the Pd-coated SnO_2_ NRs were exposed to H_2_, the H_2_ molecules decomposed into hydrogen atoms (Equation (2)). The dissociated hydrogen atoms change Pd into PdH_x_ [44], leading to a reduction in the work function of Pd (Φ_Pd_ > Φ_PdHx_) and a decrease in the height of the Schottky barrier. In addition, the hydrogen atoms reacted with a large number of the O^−^ ions (Equation (3)) that were formed at the initial state of the surface of the sample. According to the reaction in Equation (3), electrons are returned to the conduction band of SnO_2_. Therefore, the thickness of the depletion layer is reduced, thereby decreasing the resistance of the Pd-SnO_2_ NR.

## 3. Results and Discussion

Figure 3 shows an image of the as-synthesized Pd-SnO_2_ NR. Figure 3a,b show the top and cross-sectional views, respectively, of the SEM images of the Pd-SnO_2_ NRs grown on the Al_2_O_3_ substrate. Figure 3c is a magnified view of Figure 3a. The SEM image (top-view) shows that the SnO_2_ NRs predominantly were clustered and located randomly. The diameters of the NR clusters were 40–200 nm, and the distances between the adjacent clustered NR ranged from 5 to 40 nm. Figure 3c shows that the clusters consisted of small grains (~40 nm). This indicated that the diameter of the SnO_2_ NR was about 40 nm. The cross-sectional-view SEM image of Figure 3b shows that all of the SnO_2_ NRs were aligned vertically with respect to the substrate, and the average height of the SnO_2_ NR was 250 nm. Figure 3d shows the EDX mapping images of the cross-sectional-view SEM image, and it shows the SnO_2_ NR on the Al_2_O_3_ substrate. The deposited Pd nanoparticles were dispersed uniformly on the top and side surfaces of the SnO_2_ NRs. 

The sensing properties of Pd-SnO_2_ NRs in the mini-GC system were tested for various H_2_ contents. Figure 4a shows the H_2_ sensing response of the Pd-SnO_2_ NRs integrated with mini-GC to 100 ppm of H_2_ as a function of operating temperature. The optimal working temperature of the sensor was 152 °C. The response increased and reached a maximum at 152 °C, after which it decreased with increasing temperature. These results were due to the competition between the slow kinetics at lower temperatures and improved desorption at higher temperatures. 

Figure 4b shows the variation with time of the sensor signal (log(*R*)) of the Pd-SnO_2_ NRs integrated with mini-GC for various H_2_ concentrations (1–100 ppm) at the optimal working temperature of 152 °C. The inset of Figure 4b shows a magnified plot of the main figure for low H_2_ concentrations (1–10 ppm). When H_2_ was present, a noticeable peak occurred at about 10 s for various concentrations of H_2_. We found that the Pd-SnO_2_ NRs integrated with mini-GC was capable of detecting 1 ppm of H_2_, as indicated in the inset of Figure 4b. As the concentration of H_2_ increased, the height of a peak became larger. The change in the resistance was attributed to the sensing reaction of Pd-SnO_2_ NRs with H_2_, as described in Figure 2. As the H_2_ concentration increased, the thickness of the depletion layer becomes thinner due to the reaction of the $O_{2}{}^{-}$ and O^−^ ions with the hydrogen atoms that decomposed on the surfaces of the SnO_2_ NRs and Pd nanoparticles. However, the change in the resistance of the Pd-SnO_2_ NRs was found to be changed without H_2_ in ambient air. Figure 4c shows the change in the sensing signal of the Pd-SnO_2_ NRs integrated with a mini-GC column for various H_2_ concentrations at 152 °C. The change in the sensing signal was defined as Δ log(*R*) according to Equation (1), and an increase was observed with an increase in H_2_ concentration for the various H_2_ concentrations (1–100 ppm) at 152 °C.

In order to investigate the H_2_ selectivity of the Pd-SnO_2_ NRs integrated with mini-GC, we conducted measurements of the sensing signals for various gases in Pd-SnO_2_ NRs and ZnO NPs integrated with mini-GC at their respective optimal operating temperatures. Figure 5a shows the sensor signals for H_2_, CO_2_, and CH_3_COCO_3_ in the ZnO NPs integrated with mini-GC. Figure 5b shows sensor signals for air, H_2_, N_2_, CO_2_, CH_4_, and CH_3_COCH_3_ in the Pd-SnO_2_ NRs integrated with mini-GC.

In Figure 5a, the sensor signals of the ZnO NPs integrated with mini-GC were obtained upon exposure to 0.5% H_2_ in synthetic air, 5% CO_2_, and 10 ppm CH_3_COCH_3_ in N_2_. The sensing measurements were performed at the optimal operating temperature of 430 °C for the ZnO NPs integrated with mini-GC (see Appendix A). The two peaks that occurred at about 13 s were observed in the sensor signals for 0.5% H_2_ and 5% CO_2_, corresponding to the detection of H_2_ and CO_2_, respectively. The curve for 10 ppm acetone in N_2_ had two peaks, one at 13 s and one at 42 s. The first and second peak represented the emission of N_2_ and acetone, respectively. The corresponding time of the second peak was consistent with the detection time of air-balanced acetone observed in our previous work [38]. In our mini-GC system, the target gas can be separated from the mixture gas due to the different strength of interaction between the gas molecules and the stationary phase in the packed column. The stationary phase in a GC column has a weak polarity. Therefore, due to their weak interaction, non-polar gases, such as H_2_, N_2,_ and CO_2_, are released rapidly from the GC column. However, CH_3_COCH_3_, which is a polar gas, was released slowly. In this regard, H_2_ is very difficult to detect selectively in the mixture of N_2_ and CO_2_ in the ZnO NPs integrated with mini-GC.

Figure 5b shows the sensing signals when the Pd-SnO_2_ NRs integrated with the mini-GC was tested at 152 °C upon exposure to various gases, such as ambient air, 50 ppm H_2_, 5% CO_2_, 99.99% N_2_, 50 ppm CH_4_, and 50 ppm CH_3_COCH_3_ in air. Only the sensor signal of 50 ppm H_2_ had a noticeably large peak at about 10 s. There was no change in the resistance of the sensor when exposed to the other gases, because an operating temperature of 152 °C is too low to detect these gases, including carbon oxides and hydrocarbons. The results implied that the Pd-SnO_2_ NRs integrated with the mini-GC had high selectivity for H_2_ at 152 °C. According to the definition of the “Δ Sensor signal”, the sensing response of the Pd-SnO_2_ NRs to 50 ppm H_2_ was estimated to be about 0.33 (Figure 5b), which was higher than the sensing response of 0.11 when the ZnO NPs were exposed to 0.5% H_2_ (Figure 5a). In addition, the hydrogen sensing peak of the ZnO NPs at 50 ppm H_2_ was difficult to distinguish from the baseline, as shown in the inset of Figure 5a. This indicates that the ZnO NPs cannot detect low hydrogen concentrations of ˂50 ppm. Therefore, the sensing performance of the Pd-SnO_2_ NRs toward H_2_ was shown to be significantly better than that of the ZnO NPs. The superior H_2_ sensing property of the Pd-SnO_2_ NRs was attributed to the synergistic effect of the catalyst and the height of the Schottky barrier, as described in Figure 2. Accordingly, the Pd-SnO_2_ NRs integrated with the mini-GC had high selectivity and sensitivity for the detection of H_2_ at the relatively-low working temperature of 152 °C. 

We investigated the H_2_ sensing performance of Pd-SnO_2_ NRs integrated with mini-GC in a humid condition. Figure 6 shows the change of the sensor’s signal for various H_2_ concentrations (1–100 ppm) at the highly humid condition of 90% relative humidity (RH) at 152 °C. All of the sensor signals for the different H_2_ concentrations had a peak at ~10 s, corresponding the detection of H_2_. The intensity of the peaks became larger as the H_2_ concentration was increased. These results were consistent with those obtained for H_2_ in dry air (Figure 4a). The inset of Figure 6 shows the Δ Sensor signal of the Pd-SnO_2_ NRs to various H_2_ concentrations in dry and humid (90% RH) air. The results indicated that the sensitivity of the Pd-SnO_2_ NRs was decreased at 90% RH, but the sensor was able to detect H_2_ concentrations as low as 5 ppm at 90% RH.

To demonstrate the potential of the Pd-SnO_2_ NRs sensor integrated with a mini-GC device for use in real-time monitoring, the performance of the manufactured sensing measurement system was evaluated using standard hydrogen gas. To determine the hydrogen content detected, the relation function between the Δ Sensor signal and H_2_ concentration (Figure 4c) was utilized. Figure 7a–f shows the analysis process for measuring the concentration of the injected standard hydrogen gas. The measurement instructions were displayed on the front panel. After pressing the start button, a test gas of 10 ppm hydrogen was injected (Figure 7b). Gas sampling was performed for 20 s, and ambient air was used as the carrier gas (Figure 7c). After an analysis time of 100 s (Figure 7d,e), the system indicated that the concentration of hydrogen detected was 10.53 ppm (Figure 7f). Figure 8a–f shows the same analysis procedure for measuring the concentration of the hydrogen gas in exhaled breath; the results indicated that 9.66 ppm hydrogen was dissolved in the breath exhaled by the tester. This is consistent with several reports [45,46,47] that fasting basal hydrogen concentration in normal people’s breath is 8–10 ppm. According to the results, our study demonstrated that the manufactured hydrogen gas analyzer, which consisted of a Pd-SnO_2_ NRs sensor and a mini-GC device, can rapidly and accurately detect hydrogen within 100 s; consequently, it should be utilized in analyzers for breath hydrogen testing.

## 4. Conclusions

We studied the sensing performance of a breath hydrogen analyzer based on a Pd-coated SnO_2_ nanorods (Pd-SnO_2_ NRs) integrated with miniaturized gas chromatography (GC) column. We found that the breath hydrogen analyzer was capable of measuring H_2_ over a wide range of concentrations (1–100 ppm), within 100 s, at an optimal operating temperature of 152 °C. We also found that the Pd-SnO_2_ NRs could detect H_2_ selectively in mixtures of interfering gases, such as CO_2_, N_2_, CH_4_, and CH_3_COCH_3_. The higher performance of the Pd-SnO_2_ NRs compare to the ZnO NPs for sensing H_2_ was explained based on the catalytic effect and the reduction in the height of the Schottky barrier. Accordingly, we have summarized the meaningful advantages of our device as follows: smaller dimension (8 × 13 × 16 cm^3^), lower cost, lower power (12 V), smaller sampling volume (1 mL), lower operating temperature (152 °C), lower detection limit (1 ppm), and detection time (10 s). This study is a first report on a prototype of a portable and compact breath-hydrogen-analyzer device, based on MOS sensors, such as Pd-SnO_2_ NRs. The potential of the device as a breath hydrogen analyzer has been demonstrated by evaluating it with breath exhaled by a real human. More importantly, the device can selectively and sensitively detect trace amounts of hydrogen mixed in various VOC gases exhaled by humans.

## Figures and Tables

**Figure 1 sensors-22-02056-f001:**
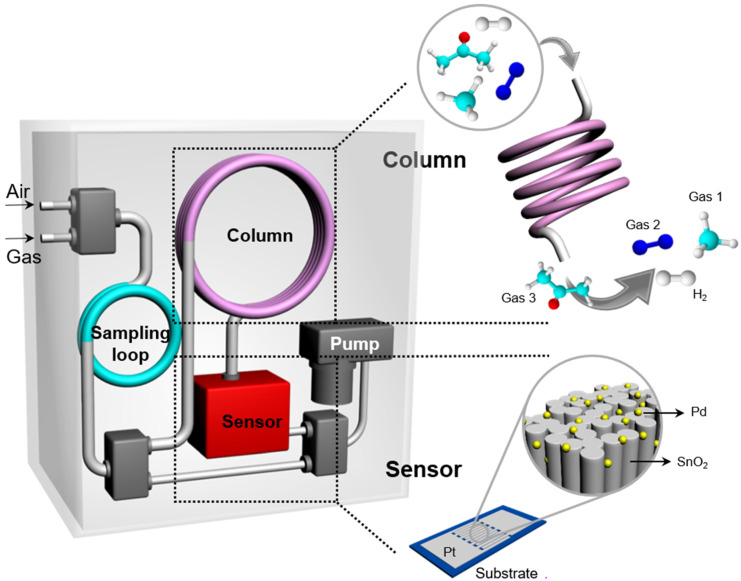
Schematic illustration of a hydrogen analyzer using a miniaturized gas chromatography (mini-GC).

**Figure 2 sensors-22-02056-f002:**
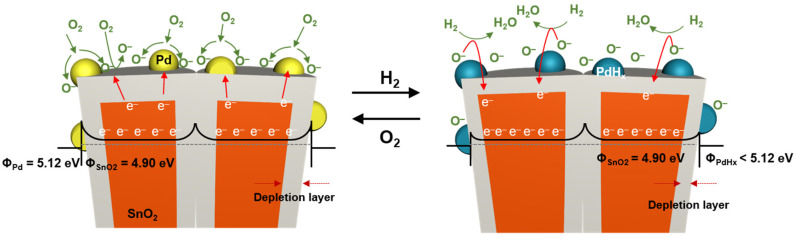
Schematic image of the sensing reaction mechanism of Pd-coated SnO_2_ NR arrays in air and hydrogen.

**Figure 3 sensors-22-02056-f003:**
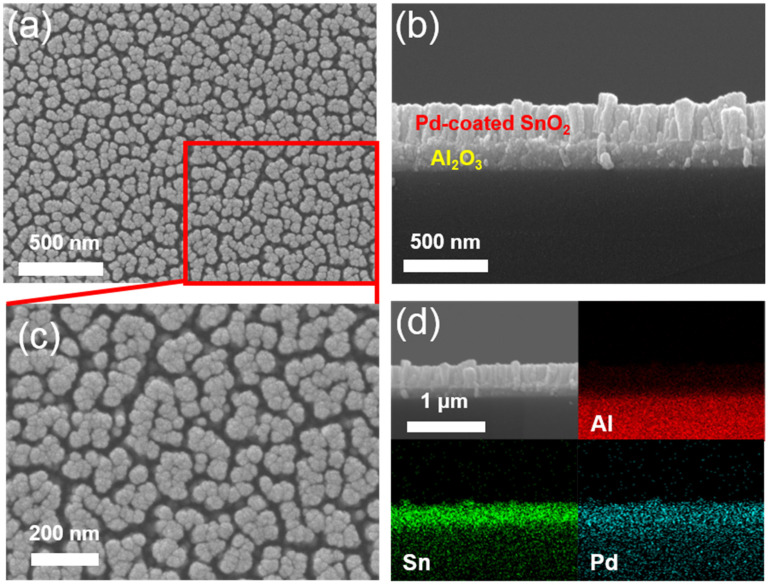
(**a**) Top-view SEM images of the Pd-coated SnO_2_ NR arrays; (**b**) cross-view SEM images of the Pd-coated SnO_2_ NR arrays; (**c**) a magnified view of (**a**); (**d**) SEM image depicting the analyzed region and the EDS elemental color mapping images for Al, Sn, and Pd.

**Figure 4 sensors-22-02056-f004:**
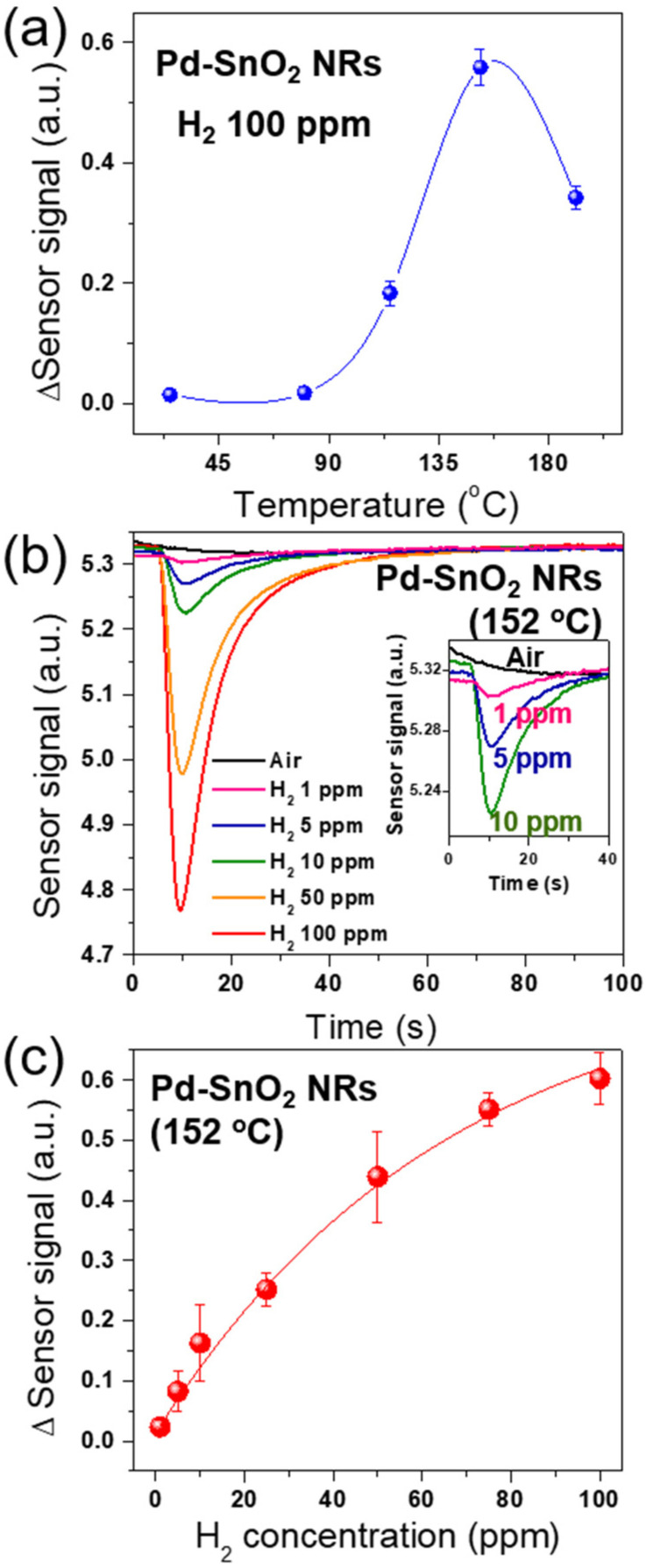
Sensing properties of the Pd-coated SnO_2_ NR arrays using the mini-GC: (**a**) sensing response (Δ Sensor signal) to 100 ppm hydrogen as a function of operating temperature; (**b**) sensor signals of the Pd-coated SnO_2_ NR arrays to various H_2_ concentrations (1–100 ppm) at 152 °C (Inset: the cases of low H_2_ concentration (1–10 ppm)); (**c**) sensing response as a function of H_2_ concentrations at 152 °C.

**Figure 5 sensors-22-02056-f005:**
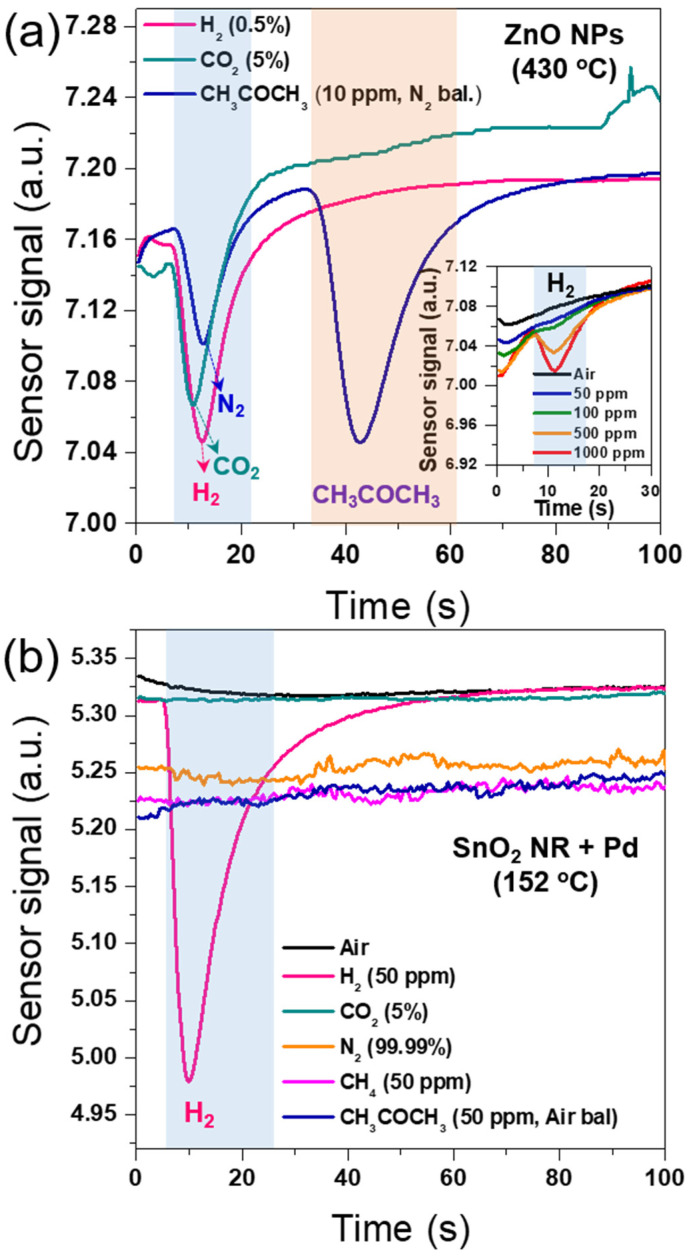
(**a**) Sensor signals for H_2_, CO_2_, and CH_3_COCO_3_ in the mini-GC integrated with ZnO nanoparticles (Inset: the cases of low H_2_ concentration (50–1000 ppm); (**b**) sensor signals for air, H_2_, N_2_, CO_2_, CH_4_, and CH_3_COCH_3_ in the mini-GC integrated with Pd-coated SnO_2_ NR arrays.

**Figure 6 sensors-22-02056-f006:**
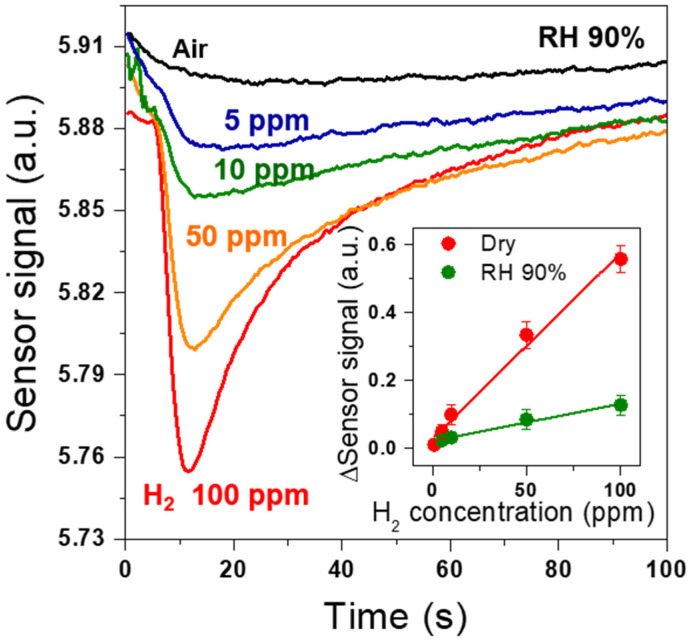
Sensor signals of Pd-coated SnO_2_ NR arrays for various H_2_ concentrations (5–100 ppm) at 90% RH and 152 °C; (inset) sensing response (Δ Sensor signal) with increasing H_2_ concentration in dry and 90% RH air.

**Figure 7 sensors-22-02056-f007:**
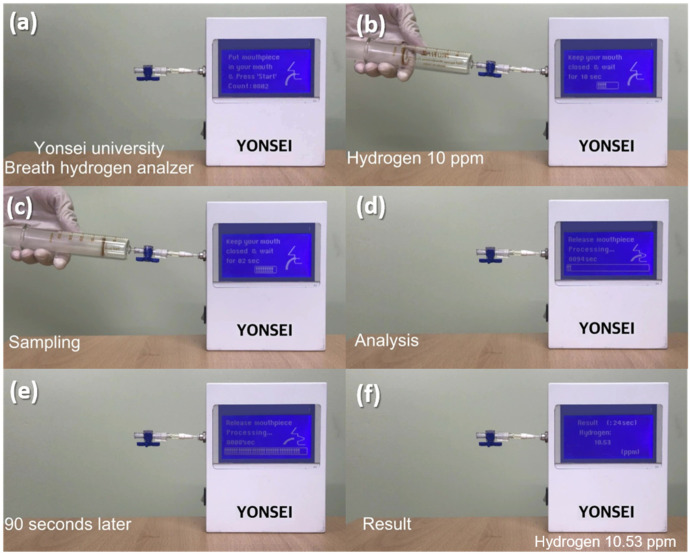
Real-time hydrogen gas sensing test via the manufactured hydrogen gas analyzer, consisting of Pd-coated SnO_2_ NR sensor and mini-GC, using 10 ppm of standard hydrogen gas. (**a**) Put mouthpiece in you mouth and Press the start button, (**b**) Injection of a standard test gas of 10 ppm hydrogen, (**c**) Gas sampling for 20 s, (**d**) Starting gas analysis, (**e**) After 90 s of gas analysis, and (**f**) Display of the gas analysis result.

**Figure 8 sensors-22-02056-f008:**
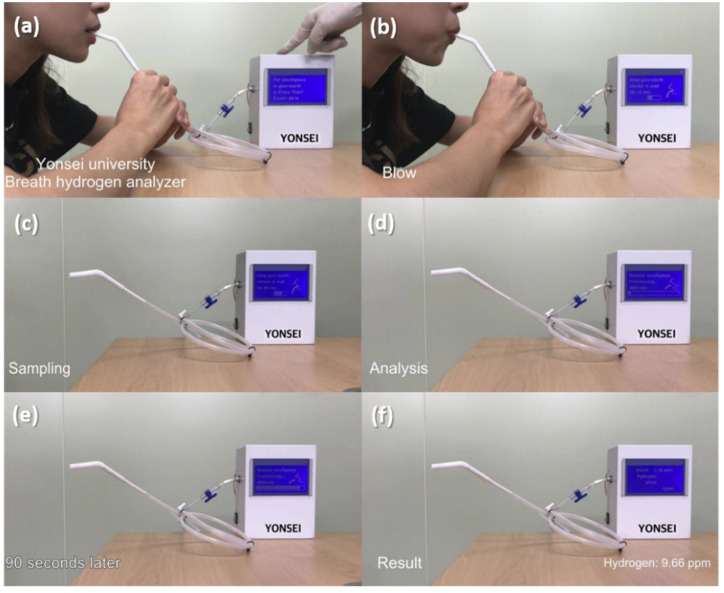
Real-time hydrogen gas sensing test via the manufactured hydrogen gas analyzer, consisting of Pd-coated SnO_2_ NR sensor and mini-GC, using tester’s exhaled breath. (**a**) Pressing the start button, (**b**) Blowing the exhaled breath for 12 s, (**c**) Gas sampling for 20 s, (**d**) Starting gas analysis, (**e**) After 90 s of gas analysis, and (**f**) Display of the gas analysis result.

## Data Availability

Not applicable.

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
