# Peer review of "Highly Sensitive and Selective Detection of Hydrogen Using Pd-Coated SnO2 Nanorod Arrays for Breath-Analyzer Applications"

_sensors, 2022, doi:10.3390/s22052056_

Round 1

Reviewer 1 Report

Comment for authors

The manuscript presents a rigorous work on the study of hydrogen detection using Pd-Coated SnO2 nanorods arrays for breath-analyzer applications. Authors did a good job with the characterization and analysis data. In overall, it is an interesting study. Following are some of my concerns:

  1. Abstract can be improved. Please mention Limit of detection (LOD).
  2. In introduction, authors briefly explain about sources of Hydrogen from human gastrointestinal. It is good to have some introduction about it, however, the main objective of this research is about the use of Metal oxide semiconductor (MOS) as sensing material. Authors should focus literature review on the previous performance of MOS as hydrogen detection. Or SnO2 detection towards other gas as comparisons. Problem statement is clear.
  1. Comparison Pd-SnO2 was compares with ZnO, however, different operating temperature were used. Are there any explanations to elaborate?
  2. Conclusion section can be improved such as the novelty of this work,
  3. References are adequate

Reviewer 2 Report

Very interesting work. Minor revision before acceptance. 1. Why Sn, specially strong Pd signal can be found at Al2O3 substrate? 2. Error bars are missing in Figure 4A. 3. Is the detection of H2 affected by other molecules? For example, can I still accurately measure H2 when I blow after eating garlic? Can you name specific substances that could affect the test results. I think that's also important. 4. The inset of Figure 6 should has the error bars as well.

Reviewer 3 Report

The paper has potential to be accepted for publication. However, I have few technical questions which needs to be addressed before the final acceptance of the paper. 

  • What is the reason to use the GLAD method? Can you please explain in the manuscript?
  • The author said “The cross-sectional-view SEM image of Fig. 3(b) shows that all of the SnO2 NRs were aligned vertically with respect to the substrate”. Related to my first question, according to my understanding, the nanorods should be tilted if GLAD method has been used. Please explain why the rods are vertical.
  • In the caption of figure 4, b is mistakenly used twice instead of c.
  • The author said “Fig. 4(d) shows sensor signals for air, H2, CO2, N2, CH4, and CH3COCH3 (acetone) in the Pd-SnO2 NRs integrated with mini-GC.” However, there is no figure 4d present in the paper.
  • In figure 5, why the gases concentration are not the same for comparison purpose? Such as CO2 99% used in figure 5a and 5% used in figure 5b.
  • In the inset of the figure 6, first of all H2 is not fully visible. Secondly “con.” Should be written as “conc.”
  • There is a little confusion in understanding, does the sensor require 10s or 100s to analyze the hydrogen gas?
  • Author has show a practical demonstration of the H2 gas sensing in figure 8. How would the author achieve 152 C temperature in the built-in device as the device is working on 12V?
  • The paper should be read thoroughly for any English Grammar mistakes. 

Round 2

Reviewer 3 Report

I am willing to accept the paper in its current form.